# Development and Visualization Improvement for the Rapid Detection of Decapod Iridescent Virus 1 (DIV1) in *Penaeus vannamei* Based on an Isothermal Recombinase Polymerase Amplification Assay

**DOI:** 10.3390/v14122752

**Published:** 2022-12-09

**Authors:** Yajin Xu, Yan Wang, Jingjie Hu, Zhenmin Bao, Mengqiang Wang

**Affiliations:** 1MOE Key Laboratory of Marine Genetics and Breeding, College of Marine Life Sciences, Ocean University of China, Qingdao 266003, China; 2Key Laboratory of Tropical Aquatic Germplasm of Hainan Province, Sanya Oceanographic Institution, Ocean University of China, Sanya 572040, China; 3Laboratory for Marine Fisheries Science and Food Production Processes, Center for Marine Molecular Biotechnology, National Laboratory for Marine Science and Technology, Qingdao 266237, China; 4Hainan Yazhou Bay Seed Laboratory, Sanya 572024, China

**Keywords:** decapod iridescent virus 1, recombinase polymerase amplification, *Penaeus vannamei*, white leg shrimp

## Abstract

Viral diseases have seriously restricted the healthy development of aquaculture, and decapod iridescent virus 1 (DIV1) has led to heavy losses in the global shrimp aquaculture industry. Due to the lack of effective treatment, early detection and regular monitoring are the most effective ways to avoid infection with DIV1. In this study, a novel real-time quantitative recombinase polymerase amplification (qRPA) assay and its instrument-free visualization improvement were described for the rapid detection of DIV1. Optimum primer pairs, suitable reaction temperatures, and probe concentrations of a DIV1-qRPA assay were screened to determine optimal reaction conditions. Then, its ability to detect DIV1 was evaluated and compared with real-time quantitative polymerase chain reactions (qPCRs). The sensitivity tests demonstrated that the limit of detection (LOD) of the DIV1-qRPA assay was 1.0 copies μL^−1^. Additionally, the presentation of the detection results was improved with SYBR Green I, and the LOD of the DIV1-RPA-SYBR Green I assay was 1.0 × 10^3^ copies μL^−1^. Both the DIV1-qRPA and DIV1-RPA-SYBR Green I assays could be performed at 42 °C within 20 min and without cross-reactivity with the following: white spot syndrome virus (WSSV), *Vibrio parahaemolyticus* associated with acute hepatopancreatic necrosis disease (*V_p_*_AHPND_), *Enterocytozoon hepatopenaei* (EHP), and infectious hypodermal and hematopoietic necrosis virus (IHHNV). In conclusion, this approach yields rapid, straightforward, and simple DIV1 diagnoses, making it potentially valuable as a reliable tool for the detection and prevention of DIV1, especially where there is a paucity of laboratory equipment.

## 1. Introduction

With a growing global population and improved nutritional awareness, the aquaculture industry has been developing rapidly [1]. *Penaeus vannamei* is presently the most important cultured shrimp in the world [2,3]. However, serious diseases affecting this species, especially the ones caused by viral infections, pose a severe threat to the global shrimp industry. Decapod iridescent virus 1 (DIV1) is a novel pathogen discovered in 2016 that has a substantial impact on the global aquaculture industry and has drawn public attention in recent years [4,5,6,7]. DIV1 is an icosahedral symmetric virus with approximately 166k bp of double-stranded DNA and has a wide host range. Susceptible species that have been reported include the following: *Fenneropenaeus chinensis*; *Macrobrachium rosenbergii*; *Procambarus clarkia*; *Penaeus monodon*; *Macrobrachium nipponense*; *Exopal aemon carinicauda*; and two species of crab, *Eriocheir sinensis* and *Pachygrapsus crassipes* [4,5,8,9,10,11]. The infected shrimp generally show an empty stomach and intestinal tract, pale hepatopancreas, and a soft shell [5]. Additionally, diseased shrimp sink to the bottom of the aquaculture pool due to their weakened swimming ability. Furthermore, dead individuals accumulate at the bottom of the aquaculture pool, and the cumulative mortality may reach 80% [12]. Because effective treatment is still unavailable for shrimp infected with DIV1, rapid and effective early pathogen detection plays a crucial role in controlling the spread of the virus in farms and reducing production losses [13].

The target sequences for the detection of DIV1 include MCP, ATPase, ribonucleotide reductase (RNR), and DNA methyltransferase genes [5,13]. Focusing on these targets, various methods have been described for the detection of DIV1, including the nested polymerase chain reaction (nested PCR), real-time quantitative PCR (qPCR), quantitative loop-mediated isothermal amplification (qLAMP), and in situ hybridization assay [12,14,15]. However, most approaches require expensive laboratory equipment, professional operation, and a relatively long amplification time. These disadvantages limit their role in rapid field detection. As a novel isothermal-amplification method, recombinase polymerase amplification (RPA) applies the complex formed by UvsX recombinase, UvsY protein, signal-stranded oligonucleotides (30–35 nt primers), and single-strand binding proteins (SSBs) to assist the site-specific D-loop strand invasion of dsDNA, and then amplifies the target DNA fragments rapidly and efficiently at ambient temperatures within less 30 min [16]. Compared to other DNA amplification methods, RPA exhibits high efficiency, rapid detection speed, simple operation, and affordable price in basic laboratory and field applications. Additionally, RPA has now been wildly used to detect bacteria, viruses, parasites, genetically modified crops, cancer, and so on [17,18,19,20,21,22,23]. Real-time quantitative recombinase polymerase amplification (qRPA), developed on the basic RPA, does not require post-amplification purification or gel electrophoresis. qRPA can analyze products quantitatively in a completely closed tube, preventing cross-contamination and false positives caused by aerosolized products [24].

In the present study, an effective real-time quantitative recombinase polymerase amplification (qRPA) assay (DIV1-qRPA assay) was described to address the current lack of rapid DIV1 field detection methods. Compared with basic RPA, the sensitivity and specificity of qRPA are higher, and qRPA does not need product purification and gel electrophoresis. The amplified products could be quantitatively analyzed in real-time with a simple fluorescence detector, which is more suitable for on-site detection [16]. Additionally, an equipment-free visual optimization of DIV1-detection results was developed in this study (DIV1-RPA-SYBR Green I assay), which is also expected to serve as novel technical guidance for the prevention and rapid on-site diagnosis of DIV1 infection.

## 2. Materials and Methods

### 2.1. Pathogen Samples and Recombinant Plasmid Construction

White spot syndrome virus (WSSV), *Vibrio parahaemolyticus* associated with acute hepatopancreatic necrosis disease (*Vp*_AHPND_), and *Enterocytozoon hepatopenaei* (EHP) were supplied by the Institute of Oceanology, Chinese Academy of Sciences. Additionally, the pathogenic liquid of infectious hypodermal and hematopoietic necrosis virus (IHHNV) was provided by the Guangxi Academy of Fishery Sciences, Nanning, China. The full length of the DIV1 ATPase gene (GenBank Accession Number: KY681040.1) was synthesized and cloned into the pUC-57 vector; then, the insert sequence was confirmed by sequencing. The recombinant plasmids were extracted by a TIANprep Mini Plasmid Kit (DP103, Tiangen, Beijing, China), and the concentration of the plasmid was then determined with a NanoDrop One spectrophotometer (Thermo Scientific, Waltham, MA, USA). The DNA copy number of the DIV1 recombinant plasmid was calculated using the following equation:$$\mathrm{DNA}\;\mathrm{copy}\;\mathrm{number}\left(\mathrm{copies}/\mu L\right)=\frac{\mathrm{concentration}\;\left(\mathrm{ng}/\mu L\right)\times 10^{-9}}{\mathrm{clone}\;\mathrm{size}\;\left(\mathrm{bp}\right)\times 660\;\left(g/\mathrm{mol}/\mathrm{bp}\right)}\times 6.022\times 10^{23}\;\left(\mathrm{copies}/\mathrm{mol}\right)$$

The recombinant plasmid was 434 ng μL^−1^, equal to 1.0 × 10^11^ copies μL^−1^, then diluted serially tenfold from 1.0 × 10^11^ to 1.0-copies μL^−1^.

### 2.2. Primer and Probe Design

Five primer pairs and probes for the DIV1-qRPA assay were designed based on the conserved regions of the ATPase gene of DIV1. These primer pairs and a probe were blasted against the NCBI nucleotide database (https://blast.ncbi.nlm.nih.gov/Blast.cgi, accessed on 18 June 2021) to ensure no homology with other organism sequences. The DIV1-qPCR assay was carried out using the previously designed primer pairs and probe [14]. All primers and probes above are listed in Table 1.

### 2.3. DIV1-qPCR and DIV1-qRPA Assay

Schematic diagram of DIV1-qRPA assay and DIV1-RPA-SYBR Green I assay is shown in Figure 1. The DIV1-qPCR reactions were performed in a 25 μL reaction mixture consisting of 12.5 μL of 2 × Premix Ex Taq (TaKaRa, Japan), 0.5 μL of PCR forward primer F_P_ (10 μM), 0.5 μL of PCR reverse primer R_P_ (10 μM), 1 μL of probe P_P_, 2 μL of DIV1 recombinant plasmids and supplementary ddH_2_O. The thermal cycling procedure included denaturation at 95 °C for 30 s and 40 cycles of amplification at 5 s at 95 °C and 30 s at 60 °C. The DIV1-qRPA reaction was executed at 35 °C for 20 min in a 50 μL volume, including 20 μL of rehydration buffer, 2.1 μL of forward primer (10 μM), 2.1 μL of reverse primer (10 μM), 0.6 μL of probe P_R_ (10 μM), 2 μL of DIV1 recombinant plasmid, 20.6 μL of ddH_2_O, and 2 μL of magnesium acetate using GenDx fluorescent kits (KS103, GenDx Biotech, Suzhou, China). To optimize the reaction conditions of DIV1-qRPA assay, seven different temperatures (30 °C, 32 °C, 35 °C, 37 °C, 40 °C, 42 °C, and 45 °C) and five different probe volumes (0.6 μL, 0.8 μL, 1.0 μL, 1.2 μL, and 1.4 μL) were screened. The DIV1-qPCR assay and DIV1-qRPA assay were carried out in a Fluorescence Quantitative PCR Detection System (CFX96, Bio-Rad, Hercules, CA, USA), and each experiment was repeated three times. All data were given as means ± standard deviation (SD) (*n* = 3). Obtained data were subjected to one-way analysis of variance (one-way ANOVA) followed by Tukey’s multiple comparisons test via GraphPad Prism 7.00, and *p* < 0.05 was considered statistically significant.

### 2.4. DIV1-RPA-SYBR Green I Assay

DIV1-RPA-SYBR Green I assay was performed using GenDx Basic kits (KS101, GenDx Biotech, Suzhou, China). The 50 µL reaction system contained 20 μL of rehydration buffer, 5 μL of best primer pair, 2 μL of DIV1 recombinant plasmid, 21 μL of ddH_2_O, and 2 μL of magnesium acetate. The reactions were performed in a VeritiPro Thermal Cycler (Thermo Fisher Scientific, Waltham, MA, USA) at optimum reaction temperature screened from the process above for 20 min. Then, 2 μL of obtained production was verified by 1.5% gel electrophoresis, and the remaining products were mixed with 2 μL of SYBR Green I nucleic acid dye (1:10 dilution of 10,000× stock solution, Solarbio, Beijing, China). Then, their fluorescence intensities were observed with a 302 nm UV. To optimize the ideal visualization conditions, five different final primer concentrations of 0.3 μM, 0.2 μM, 0.1 μM, 0.05 μM, and 0.025 μM were designed to avoid the false positive caused by the dimer.

### 2.5. Evaluation of Sensitivity and Specificity

To compare the sensitivity among DIV1-qPCR, DIV1-qRPA, and DIV1-RPA-SYBR Green I assay, the gradient diluted DIV1 plasmid sample with a concentration from 1.0 × 10^5^ to 1.0 copies μL^−1^ was used as a template in the positive control group and the equivalent volume of ddH_2_O was used as template in the no-template control (NTC) group to estimate the LOD of the three assays. Standard curves of DIV1-qPCR and DIV1-qRPA were created versus the concentration gradient of the diluted plasmid, and their Ct values were calculated and expressed as mean ± SD.

To determine the specificities of DIV1-qRPA and DIV1-RPA-SYBR Green I assay, the DIV1 plasmid sample with a concentration of 1.0 × 10^5^ copies μL^−1^ and the other DNA samples for the specificity assays containing WSSV, IHHNV, EHP, and *Vp*_AHPND_ were used as templates in the positive control group, and the equivalent volume of ddH_2_O was used as a template in the NTC group for experiments.

## 3. Results

### 3.1. Primer Screening

To optimize subsequent reaction conditions, five different sets of primers were screened based on the threshold time. According to the amplification curve after reacting at 35 °C for 20 min, all five sets of primers showed good amplification efficiency. All primer pairs reached the detection threshold within 7 min and showed a plateau period after that. Even the lowest end-point relative fluorescence unit (End-RFU) was almost higher than 1000 (Figure 1A). Among all the primer pairs, the fifth group of primers (F_R5_ and R_R5_) first reached the detection threshold less than 4 min after the reaction began, and the End-RFU was significantly higher than that of other groups, at 2385.5 ± 64.6 (Figure 1A,B). Therefore, the fifth set of primers was selected for use in the following assays.

### 3.2. Optimizing the Reaction Temperature of DIV1-qRPA Assay

To improve the conditions for subsequent assays, seven different temperatures (30 °C, 32 °C, 35 °C, 37 °C, 40 °C, 42 °C, and 45 °C) were screened based on the End-RFU and the threshold time. For all the tested temperatures, the target fragments were effectively amplified, reached the detection threshold within seven minutes, and showed a subsequent plateau period (Figure 2A). Particularly, the amplification curve at 42 °C first reached the detection threshold at 1.14 ± 0.146 min, *p* < 0.05, and arrived at the platform stage at the 8th minute (Figure 2B). Meanwhile, a relatively high average End-RFU with a value of 3277.2 ± 143.9, *p* < 0.05, was observed at 42 °C, really close to the maximum End-RFU of 3557.6 ± 137.8, *p* < 0.05, at 40 °C (Figure 2C). Thus, 20 min and 42 °C were chosen as the subsequent testing reaction conditions.

In the range of 37–45 °C, the threshold time was relatively shorter than other groups (Figure 2B). When the set temperature was lower than 42 °C, the threshold time was shortened as the reaction temperature gradually increased. However, when the reaction temperature reached 45 °C, the detection threshold time suddenly lagged behind that at 40 °C. An interesting finding is that the End-RFU obtained at 37–42 °C was comparatively higher than other sets, and the End-RFU of 45 °C (2006.2 ± 131.7) was the lowest among all the setting temperatures (Figure 2C). Hence, by taking the intersection of these two temperature ranges, we concluded that the best temperature range for the DIV1-qRPA assay is 37–42 °C.

### 3.3. Optimizing the Probe Consumption of the DIV1-qRPA Assay

To optimize the probe consumption, four different probe volumes (0.6 μL, 0.9 μL, 1.2 μL, and 1.8 μL) with a concentration of 10 μM were respectively added to the reaction system. All groups were successfully amplified except the NTC group (Figure 3A). When the probe volume was no more than 1.2 μL, the corresponding End-RFU increased with the increase in probe volume, and the End-RFU of 1.4 μL (3038 ± 245.0) was similar to that of 1.2 μL with 3267.5 ± 159.0, *p* < 0.05 (Figure 3B). Hence, we finally utilized 1.2 μL of the 10 μM probe in the subsequent sensitivity experiments.

### 3.4. Sensitivity Evaluation of qPCR and qRPA Assays

To test the sensitivity, six sets of gradient-diluted DIV1 plasmid samples with concentrations from 1.0 × 10^5^ to 1.0 copies μL^−1^ and the equivalent volume of ddH_2_O were used to analyze the DIV1-qPCR, DIV1-qRPA and DIV1-RPA-SYBR Green I assays. Analyzing the amplification curve in the DIV1-qRPA assay, we observed that the six sets were all successfully amplified except for NTC, and all sets reached the detection threshold in 10 min (Figure 4A).

The standard curve showed that the DIV1-qRPA assay had a high correlation coefficient (R^2^ = 0.9891) within the range of 1.0 × 10^5^–1.0 DNA copies μL^−1^ (Figure 4B). The regression equation was Ct=−3.517logn+21.34 (n = DIV1 DNA copies). In the meantime, the standard curve of the DIV1-qRPA assay also showed a high correlation coefficient (R^2^ = 0.9985) within the range of 1.0 × 10^7^–1.0 × 10^1^ DNA copies μL^−1^ (Figure 5A), and the regression equation of qPCR was Ct=−3.221logn+39.78 (n = DIV1 DNA copies). Compared with the DIV1-qPCR assay, the DIV1-qRPA assay has a more stable amplification effect at a low input template concentration (Figure 5B), and the LOD of the DIV1-qRPA assay was 1.0 copies μL^−1^ higher than the LOD of the qPCR (10 copies μL^−1^).

### 3.5. Optimizing the Primer Concentrations of DIV1-RPA-SYBR Green I Assay

To prevent false positives caused by dimers, five alternative final primer concentrations (0.3 μM, 0.2 μM, 0.1 μM, 0.05 μM, and 0.025 μM) were set to select the optimum concentration based on the fluorescence results and electrophoresis images. The fluorescence results indicated that when the primer concentration was higher than 0.025 μM, the difference between the NTC and the positive sample was too small to distinguish between them, while the NTC had the lowest background value at 0.025 μM and its dimer was also the lightest in the corresponding electrophoresis image (Figure 6A). Therefore, 0.025 μM was selected as the optimal primer concentration for DIV1-RPA-SYBR Green I.

### 3.6. Sensitivity Evaluation of RPA-SYBR Green I Assay

The sensitivity of the DIV1-RPA-SYBR Green I assay was determined by observing the fluorescence results of each tube after the reaction was completed. The results show that reducing the primer concentration indeed affected the detection limit, and the sensitivity of the DIV1-RPA-SYBR Green I assay was still high. When the input template concentration was lower than 1.0 × 10^3^ copies μL^−1^, there was little difference between the positive and NTC samples to the naked eye under UV light, indicating the LOD of the DIV1-RPA-SYBR Green I assay was 1.0 × 10^3^ copies μL^−1^ (Figure 6B).

### 3.7. Specificity Evaluation of qRPA and RPA-SYBR Green I Assay

To test the specificity, the DIV1 plasmid sample with a concentration of 1.0 × 10^5^ copies μL^−1^, the ddH_2_O, and the DNA templates of WSSV, IHHNV, EHP, and *Vp*_AHPND_ were used to determine the DIV1-qRPA assay and the DIV1-RPA-SYBR Green I assay. The amplification curves or green fluorescence were only observed from positive plasmid samples, suggesting that the DIV1-qRPA assay and DIV1-RPA-SYBR Green I assay both had good specificity (Figure 4C and Figure 6C).

## 4. Discussion

DIV1 is a recently identified pathogen in crustaceans and poses a severe threat to the aquatic industry in China and around the world [8]. Because the onset of DIV1 is currently difficult to treat effectively, it is, therefore, urgent to develop a rapid and sensitive field detection method to prevent this virus in advance [13,15]. RPA, as a newly emerged isothermal molecular detection method, has gained popularity in pathogen detection, especially since it is rapid, simple, sensitive, specific, and affordable to use. In the present study, the novel DIV1-qRPA assay and the DIV1-RPA-SYBR Green I assay we developed could rapidly detect DIV1 with high sensitivity and specificity and are suitable for field detection.

Previous studies have shown that the RPA reaction could operate at temperatures ranging from 22 °C to 45 °C [25]. In this study, we established that the optimal temperature range for the RPA reaction was 37–42 °C based on the End-RFU and the threshold time. The amplification efficiency at all temperatures of this range was good. This result was consistent with previous studies showing that the RPA reaction does not require precise temperature control in this range [25]. Moreover, from the abnormally low End-RFU of 45 °C (2006.2 ± 131.7) (Figure 2C), we inferred that a high reaction temperature might lead to the inactivation of enzymes in the system and thus greatly reduce the amplification efficiency of RPA. In addition, it was also proven that the RPA reaction could be carried out at body temperature, so the requirement for external heating equipment might be reduced [19]. According to a previous report, the threshold fluorescence value could be reached within 5–8 min with agitation beginning at the fourth minute of the reaction; otherwise, the time it takes to reach a detectable level without agitation is between 8 and 14 min [26]. In our operation, a stable amplification curve was more likely to form when agitation was fully performed at the beginning of the process and the third or fourth minute of the reaction. Moreover, when the concentration of the input template was low, the LOD might be improved by taking out the reaction mixture and fully mixing it several times in the fluorescence measurement gap. In addition, the LOD could also be improved by appropriately increasing probe consumption when the template input concentration was low.

Compared with the DIV1-qPCR assay, the DIV1-qRPA assay saved almost half of the DIV1-qPCR’s reaction time and obtained a higher sensitivity of 1.0 copies μL^−1^. The LOD of the DIV1-qPCR was also higher than that of all current detection methods for DIV1. Although the qLAMP assay could be performed at a constant temperature too, the six intricate oligonucleotide primers, a hot reaction temperature of 60 °C, and a lengthy reaction period of 45 min were necessary [27]. In addition, it has been reported that the repeatability of the qLAMP assay could be poor when template concentration was lower than 10^3^ DNA copies μL^−1^ [14]. Compared with the qLAMP, the DIV1-qRPA assay required a lower reaction temperature and shorter amplification time and showed a more stable amplification effect at a low input template concentration. Therefore, the DIV1-qRPA assay could not only be used as an alternative method for detection in the laboratory, but it is also suitable for rapid detection in the field with simple equipment.

RPA products can be equipment-free and visually analyzed using a variety of methods. In addition to the qRPA method mentioned in this study, the recombinase polymerase amplification and a lateral flow dipstick (RPA-LFD) method are also frequently used to detect pathogens [28,29]. However, a drawback of using the RPA-LFD test for the identification of pathogens is the potential post-amplification contamination of samples in field settings [24]. Moreover, the use of gold nanoparticles, fluorescence-labeled probes, biotin, biotin–ligand complexes, and antibodies has resulted in a large increase in the cost of evaluating high-throughput clinical samples. Hence, we employed the SYBR Green I to optimize an affordable visual analysis method for DIV1-qRPA detection results that eliminated the risk of potential sample contamination. The DIV1-RPA-SYBR Green I assay could also maintain a high level of specificity and sensitivity that achieved a LOD of 1.0 × 10^3^ copies μL^−1^. This is the first study to develop a DIV1-RPA-SYBR Green I assay for rapid and sensitive DIV1 detection. The potential flaw of the RPA-SYBR Green I assay for DIV1 is that SYBR Green I dye can nonspecifically bind to any double-stranded DNA. Therefore, when there is a lot of template DNA or primers present, the specificity of DIV1 detection would be decreased. Previous studies have attempted to limit the amount of DNA templates or primers in the reaction system to ensure the specificity of RPA-SYBR Green I assay [30,31]. In this study, we optimized the concentration of primers in the reaction system with the other conditions fixed and also assisted gel electrophoresis results in avoiding subjective judgment. False positives might also occur with high DNA content, and false negatives could conversely occur with low DNA content. A total amount of 300 ng to 2 μg of DNA template in a 50 μL reaction system has been recommended to avoid false negative and positive detection results in clinical testing in the field [30]. In this study, false positives due to primer dimers were not observed when 2 μL templates with concentrations of 1.0 × 10^5^ copies μL^−1^ or less were added to the mixture system at a final primer concentration of 0.025 μM. Moreover, low-cost commercial nucleic acid extraction methods for field samples, such as magnetic bead-based technology and heated NaOH method, could be used to further reduce the cost of RPA detection [32].

## 5. Conclusions

We developed a highly sensitive and specific real-time quantitative RPA assay and improved its instrument-free visualization for rapidly detecting DIV1. The LOD of the DIV1-qRPA assay reached 1.0 copies μL^−1^, which was higher than the LOD of qPCR and qLAMP, and the visual detecting limitation of the instrument-free DIV1-RPA-SYBR Green I assay was 1.0 × 10^3^ copies μL^−1^. Both assays could be performed at 42 °C within 20 min and had no cross-reactivity with WSSV, *Vp*_AHPND_, EHP, or IHHNV. These two methods offer straightforward, eye-catching, and equipment-free approaches for DIV1 detection in shrimp farms, quarantine stations, and basic laboratories with limited resources, especially in remote and rural regions; the most appropriate method can be chosen based on the practical conditions of the testing site. Furthermore, the results of this study may promote the wide application of DIV1 detection methods based on nucleic acid amplification technology and provide a reference value for monitoring and controlling this new virus in the aquaculture industry.

## Data Availability

Not applicable.

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
