# Peer review of "Development and Visualization Improvement for the Rapid Detection of Decapod Iridescent Virus 1 (DIV1) in Penaeus vannamei Based on an Isothermal Recombinase Polymerase Amplification Assay"

_viruses, 2022, doi:10.3390/v14122752_

Round 1

Reviewer 1 Report

Overall, this is a well-presented paper describing a comprehensive evaluation of a novel assay for decapod iridescent virus 1 (DIV1). I enjoyed reading this work and was impressed by the detailed analysis that was undertaken. There are several typographical errors in the discussion, which I have listed below. I also urge you to read through the discussion yourself and evaluate your sentence structure and word usage.

Ln256 “rapid” should be “rapidly”

Ln257 “were” should be “are”

Ln259 “exploded” in not the right word in this sentence, change to “found” or “established” or something similar

Ln 260-263 sentence “The amplification efficiency…….with previous research” is too long. You could put a full-stop at “good” and start a new sentence after this.

Ln287 “visual” should be “visually”

Ln 295-295 “afforded” should be “affordable”

Ln301-302 “primer sin” should be primers in”

Ln 305 “fixe should be “fixed”

Reviewer 2 Report

  In this study, authors descrivbes " Development and visualization improvement for the rapid detection of the decapod iridescent virus 1 in Penaeus vannamei based on an isothermal recombinase polymerase amplification assay ". The method of rapid detection of WSSV detection using LAMP has being used in double Spiral Co. Guangzhou, China. What’s the novel method or idea in this Ms?

1.     The detection of the different viral strains should be added using this method.

2.     Error bars were the independent experiments, or the replication of one sample?

3.     statistical analysis should be added in Figures.

Reviewer 3 Report

This manuscript presents an application of Development and visualization improvement for the rapid detection of the decapod iridescent virus 1 in Litopenaeus Vannamei based on an isothermal recombinase polymerase amplification assay. lt is a topic of interest to the researchers in the related areas but the paper needs very significant improvement before acceptance for publication. My detailed comments are as follows (Note that each number corresponds to a line number):

43. Eliminate multiple references. After that please check the manuscript thoroughly and eliminate all the lumps in the manuscript. This should be done by characterizing each reference individually. This can be done by mentioning 1or 2 phrases per reference to show how it is different from the others and why it deserves mentioning. Reference can be made to other published research literature in the journal.

54. Adjacent literature citations should be [13,14].

65. Same as above 42.

70. The assay can be performed in combination with clinical samples, thus evaluating the practical application of the method for detecting Decapod iridescent virus 1 (DIV1) in this study.

88. The registration number corresponding to the conserved regions ATPase gene of DIV1.

89. The content involving open source sites need to write the corresponding site.

92. Check all format, e.g. FP in the qPCR column of Table 1, P should be in subscript, check all.

93. The title of DIV1-qPCR and DIV1-qRPA assay does not match the title on line 113.

95. Please use SI unit. Including those in figures/tables and leave a space between the value and unit. Please check all.

97. Same as above 92

113. Graphical abstract (GA) should be a high-quality illustration or diagram in any of the following formats: PNG, JPEG, TIFF, or SVG. The minimum required size for the GA is 560 pixels×1100 pixels(height×width),we recommend a minimum resolution 600dpi. When submitting larger imagesthe size should be of high quality in order to be easily reproducible. Please check all GA.

117. The sentence “best primer pair selected from above of optimum concentrations” is not clearly expressed can be written directly to select the best primer pair.

141. The initial concentration of plasmid samples is not mentioned.

146. Check all format, e.g. (Fig.1A) in the article should be complete (Figure 1A), check all.

151. Same as above 113.

160. P<0.05, P should be italicized, check elsewhere in the article.

168. “At the same time” this sentence is somewhat raw in expression and can be expressed as an interesting finding is…

171. The inferences can be put in the discussion section

176. Same as above 113.

188. Same as above 113.

188. Rephrasing the explanatory information in Figure 3, it should not be the relationship between probe concentration and other factors, but should be expressed as the relationship corresponding to the amount of different probes used.

206. There is a problem with the expression, the logic of comparing oneself to oneself is not correct.

210. Same as above 113.

218. Same as above 113.

241. Same as above 113.

251. The discussion section on Decapod iridescent virus 1 (DIV1) can be appropriately expanded to describe the content.

252. The transition to RPA is a bit abrupt here, and it is suggested to write that the onset of this virus is difficult to treat, so it is important to detect the virus to prevent it in advance. List the relevant assays and cite the literature.

290. Same as above 54.

303. Same as above 54.

341. Consult the journal's reference style for the exact appearance of these elements, and use of punctuation and capitalization.The format of the reference should refer to the “Information For Authors” of the journal.

Reviewer 4 Report

The manuscript ID viruses-2045190, titled: “Development and visualization improvement for the rapid detection of the decapod iridescent virus 1 in Penaeus vannamei based on an isothermal recombinase polymerase amplification assay”. Minor revisions are important to be addressed before going to accept this article.

1.    Correct the title to “virus 1 (DIV1) in Penaeus vannamei”, not all readers know the meaning of virus 1 P. vannamei. Please, correct this to be easier for readers.

2.    In Keywords, White Leg shrimp should be added

3.    The introduction needs further improvements to introduce the point of study well. Authors may use the following refs:

https://doi.org/10.1111/are.14566

https://doi.org/10.1016/j.aquaculture.2022.737905 

https://doi.org/10.3389/fphys.2022.874172

4.    Improve the quality of Figs.

5.    At first time, the scientific name should be full-written, then write the abbreviation

6.    Line 320: Correct “μL−1” to “μL−1”. Also, revise these in all the manuscript

7.    The conclusions need further improvements. It should build on what is already there to say something about the significance of the findings for existing research, future research, and study implications.

Best Regards

Round 2

Reviewer 2 Report

The revised Ms can be published